# Laser Additive Manufacturing of TC4/AlSi12 Bimetallic Structure via Nb Interlayer

**DOI:** 10.3390/ma15249071

**Published:** 2022-12-19

**Authors:** Zhicheng Jing, Xiangyu Liu, Wenbo Wang, Nuo Xu, Guojian Xu, Fei Xing

**Affiliations:** 1School of Material Science and Engineering, Shenyang University of Technology, Shenyang 110870, China; 2School of Mechanical Engineering, Shenyang University of Technology, Shenyang 110870, China; 3Shenyang Zhongke Raycham Technology Co., Ltd., Shenyang 110870, China; 4Department of Development, Reform and Discipline Construction, Shenyang University of Technology, Shenyang 110870, China

**Keywords:** laser additive manufacturing, TC4/AlSi12 bimetallic structure, Nb interlayer, intermetallic compounds

## Abstract

The TC4/AlSi12 bimetallic structures (BS) with Nb interlayer transition were fabricated by laser additive manufacturing (LAM). The results showed that the TC4/AlSi12 BS with Nb interlayer prepared with optimized process parameters can be divided into three regions (the TC4 region, Nb region and the AlSi12 region) and two interfaces (the TC4/Nb interface and the Nb/AlSi12 interface). The high melting point (Ti, Nb) solid solution formed in the Nb region acted as a diffusion barrier between the TC4 alloy and the AlSi12 alloy, thereby effectively inhibiting the formation of Ti-Al intermetallic compounds (IMCs). With the decrease of the laser output power for AlSi12 deposition, the NbAl_3_ IMC changed from layered to dispersed distribution, while γ-TiAl and Ti_5_Si_3_ IMC disappeared, thus significantly reducing the crack susceptibility of the BS deposited layer. The tensile strength of TC4/AlSi12 BS with Nb interlayer was about 128MPa, and the fracture was located near the Nb/AlSi12 interface.

## 1. Introduction

The increasingly complex working conditions in aerospace engineering have necessitated higher requirements for the design and service performance of various structures, making metal materials face new challenges in structural design and manufacturing. Multi-level, lightweight and low-cost design and manufacturing have become a research hotspot for structural materials [1,2]. In view of this, the development and application of composite structures in the aerospace field has been promoted. The Ti alloy/Al alloy bimetallic structure (BS) is one of the structures most studied by scholars. Ti alloy/Al alloy BS combines the advantages of low density and good economy of the Al alloy and the high strength and good corrosion resistance of the Ti alloy. While giving full play to their respective advantages, it can effectively reduce the structure weight and manufacturing cost. The design concept of replacing titanium alloy integral structural parts with Ti/Al composite structures has long been proposed to unify economy and reliability into one component [3]. Currently, Ti alloy/aluminum alloy BS components have been reported to be used in fighter aircraft wings, automobile exhaust systems and track seats of Airbus aircraft to improve local corrosion resistance along with lightweight and high specific strength [4,5,6]. Therefore, it has broad application prospects in the fields of aerospace, weapon equipment and transportation.

In the past decades, scholars have mainly adopted traditional dissimilar metal welding (laser welding, electron beam welding, diffusion welding, friction stir welding, surfacing welding and explosion welding, etc.) and spraying methods to obtain high-performance Ti alloy/Al alloy dissimilar metal structural materials. However, the metallurgical reaction between Ti and Al produces a variety of -intermetallic compounds (IMCs) during welding, including Ti_3_Al, TiAl, and TiAl_3_, etc., which reduces the ductility and strength of the weld and are the main reason for the formation and propagation of crack sources [7,8]. In order to suppress the formation of IMCs between Ti-Al and reduce the crack sensitivity of welded joints, an Al-Si wire was used to braze the Al alloy and Ti alloy. The results showed that a thin layer of ternary phase (Ti_7_Al_5_Si_12_) formed before TiAl_3_ due to the segregation of Si atoms near the bonding surface of the solid Ti alloy, effectively inhibiting the formation of TiAl_3_ [9,10]. In addition to controlling the weld pool composition by filler metal to reduce Ti-Al IMCs, reducing welding heat input and increasing weld cooling rate can also result in a thinner IMC layer at the Ti alloy/Al alloy joints. Zhang et al. [11] proposed a new method in which a layer of pure titanium mesh was placed between the Ti6Al4V and A6061 plates as an interlayer. This allows the molten aluminum alloy to have sound diffusion and wetting ability on the surface of the titanium alloy even at the lower welding heat input, ensuring sufficient strength of the brazing joint. In conclusion, no matter which welding method was used, it was an attempt to reduce the number and layer thickness of Ti-Al IMCs in order to improve the mechanical properties of the joint. In addition to the generation of various IMCs, the significant differences in thermal expansion coefficient and thermal conductivity between Ti alloy and Al alloy lead to welding residual stress and improve the crack sensitivity of the welded joints. In addition to welding, Naeimian et al. [12] studied the microstructure and mechanical properties of diffusion-bonded Al-Ti joints using Babbitt thermal spray coat as the interlayer. Charles et al. [13] used cold gas-dynamic spray to prepare and characterize titanium coatings of 3 mm thickness on Al 6063 substrate, although the number of Ti-Al IMCs can be reduced to a certain extent by spraying. However, it has some limitations and is only suitable for surface modification or preparation of Ti alloy and Al alloy dissimilar metals with a simple structure and small size. In summary, the direct joining of Ti alloy and Al alloy still faces severe challenges.

The emergence and development of Additive Manufacturing (AM) technology provide a new manufacturing method for connecting dissimilar materials (BS materials). Laser Additive Manufacturing (LAM), also known as Laser Engineered Net Shaping (LEN) or Laser Melting Deposition (LMD) is one of the encouraging additive manufacturing methods. It uses the laser as forming heat source and has the characteristics of high process applicability and forming efficiency. It breaks through the technical barrier whereby dissimilar welding and spraying methods are only suitable for thin plate structural parts, and can meet the manufacturing requirements of increasingly complex structural parts, especially large ones. The direct, gradient transition or interlayer transition fabrication of BS can be achieved by LAM. For example, Heer et al. [14] prepared SS430 and SS316 magneto-nonmagnetic bimetallic gradient structure materials using AM technology, which showed that different characteristic zones could be created in specific areas of a part by LAM technology alone, without the traditional joining steps. Onuike et al. [15] successfully produced crack-free Inconel718/Ti64 BS by LEN technology using CBL (a mixture composed of VC, Inconel 718 and Ti64) as an intermediate bonding layer. Ma et al. [16] successfully prepared TC4/TiAl bimetallic gradient structure materials by LMD technology, which reduced the difference in thermal expansion coefficient of the two materials and eliminated cracks in the deposition layers.

In recent years, the research of fabricating BS using interlayer metal transition has attracted much attention from scholars. The selection principle of the interlayer metal is to form no or less IMCs with the two main metals to reduce crack sensitivity during the metallurgical reaction. It can be seen from the phase diagram of Ti-Nb binary alloy that Nb and Ti can form an infinite solid solution. Gao et al. [17] have confirmed through experiments that no IMC brittle phase was formed during the welding process of Ti6Al4V/Nb dissimilar alloys. Gao et al. [18] avoided the formation of a Ti_x_Cu_y_ brittle phase by preventing the mixing of Ti and Cu in the melt pool through the Nb interlayer and successfully prepared Ti6Al4V/Cu welded joints. Numerous research results have shown that the proper selection of interlayer can effectively prevent the mutual diffusion and reaction between the two base materials. According to the Al-Nb binary alloy phase diagram, only NbAl_3_ IMC can be formed between Al and Nb in an extensive composition range (wt.%Nb ≤ 53%Nb). Moreover, due to the significant difference in melting point between Nb (2468 °C) and Al (660 °C), controlling the interlayer temperature through appropriate heat input adjustment can reduce the melting content of Nb in the liquid molten pool, thereby effectively suppressing the formation of Nb-Al IMCs. Therefore, it can be considered that Nb is an ideal interlayer metal for the preparation of TC4/AlSi12 BS.

In this study, LAM technology was used to prepare TC4/AlSi12 BS samples through the Nb interlayer under two different process parameters. The study focused on adjusting the microstructure of the BS by precisely controlling the power of the deposited layer to obtain high-quality Ti alloy/Al alloy BS free of cracks and other metallurgical defects. The microstructure morphology, phase transformation and the crack elimination mechanism of the TC4/AlSi12 BS deposition layer under two parameters were analyzed in detail.

## 2. Materials and Methods

### 2.1. Materials Preparation

TC4, AlSi12 and Nb (purity 99.99 at%) metal powders were provided by (Beijing) Jinggao Excellent Materials Co., Ltd. (Beijing, China). All three powders were prepared by rotating electrode technology with a particle diameter in the 75–150 μm range. The chemical compositions of TC4 and AlSi12 powders are shown in Table 1. Before the experiment, TC4, AlSi12 and Nb powders were heated and dried in a vacuum drying oven at 100 °C for 4 h to eliminate the moisture absorbed on the surface of the powders. The hot-rolled TC4 titanium alloy plate with the size of 200 × 100 × 20 mm was used as the experimental substrate. Before deposition, the substrate was mechanically polished to remove the oxide film and defects on the surface, and then wiped with acetone and blown dry to remove impurities such as oil stains.

### 2.2. The LAM Process of TC4/AlSi12 BS

The TC4/AlSi12 BS were performed on the LDM8060 system developed by Nanjing Zhongke Raycham Laser Technology Co., Ltd., (Nanjing, China), as shown in Figure 1a. The LAM experiment was carried out in an argon atmosphere and the oxygen and steam content in the inert gas chamber was strictly controlled below 20 ppm. The deposition sequence of TC4/AlSi12 BS was to first deposit a TC4 cuboid with a size of about 40 × 25 × 20 mm on the TC4 substrate. -Secondly, a layer of Nb was deposited on top of the TC4 deposit. An AlSi12 cuboid of about 40 × 25 × 20 mm was then deposited on the Nb interlayer, and finally BS with a total size of about 40 × 25 × 40 mm was deposited, as shown in Figure 1a. The process parameters for samples 1 and 2 are shown in Table 2, which were obtained on the basis of extensive preliminary experiments [19]. The physical photos of the samples are shown in Figure 1c,d; compared with sample 1, the reduction of laser power during the deposition of AlSi12 in sample 2 leads to a significant alleviation of its collapse, resulting in a flatter and smoother surface and sides.

### 2.3. Characterization Methods

After LAM, the metallographic specimen for analysis was ground with #200–#3000 sandpaper and mechanically polished with SiO_2_ suspension liquid. Next, Keller reagent (HNO_3_:HCL:HF:H_2_O = 2.5:1.5:1:95) was used to etched the metallographic specimens. The microstructure of the BS cross-section (YOZ plane) was observed by ZEISS Gemini SEM 300 field emission scanning electron microscope (SEM). Accurate phase composition and crystallographic analysis of sample 1 were carried out by Electron Backscattered Diffraction (EBSD) equipped with Gemini SEM 300 field emission SEM. The surfaces used for EBSD testing were mechanically polished for 3 h and ion polished at 6.5 kV for 30 min. The acquisition parameters were the working voltage of 20 kV, the speed of 22.18 Hz, the inclination angle of 70°, and the step size of 0.6 μm. A micro-area X-ray diffraction analysis (XRD) was performed to determine the phase composition of the pure TC4 region, pure AlSi12 region and Nb region in sample 2, with a copper target (λ = 1.5406) and a scan range of 20°~90° at a scan speed of 4°/min. The Vickers hardness distribution of BS was tested by a Wilson Hardness UH 250 multi-functional hardness tester, with a load of 9.8 N and a dwell time of 10 s. The tensile properties of the samples were tested by a CSS-55100 universal testing machine at a tensile loading rate of 0.2 mm/min. The tensile specimens were cut by wire cutting with the geometry and dimensions shown in Figure 1b, and the Nb deposit interlayer was ensured to be in the middle of the specimen. After a tensile test, the morphology and composition of tensile fracture were analyzed by a Hitachi S-3400 SEM and its equipped EDS.

## 3. Results and Discussion

### 3.1. Microstructure of the Cross-Section

The microstructure of sample 1 and 2 is shown in Figure 2. Four regions and three interfaces were formed in sample 1 (Figure 2a), which were named as the TC4 region, Nb region, IMC region and AlSi12 region, respectively. In sample 1, the deposited Nb powders were completely melted and formed the layered Nb region with a thickness of about 850 μm. An IMG layer with a width of about 600 μm was then formed between the Nb region and the AlSi12 region due to the diffusion behavior of elements. A through-crack starting from the IMC/AlSi12 interface and ending at the Nb/IMC interface was found in the IMC region. It should be noted that the initiation (Figure 2b) of the crack appears to be in a healed state filled with Al alloy. However, due to the rapid cooling of the LAM process, the liquid Al alloy with good fluidity did not have time to heal the entire crack (Figure 2c), which destroyed the continuity of the organization. In order to suppress the metallurgical reaction between Nb and Al and avoid the generation of IMC layer and crack, sample 2 was prepared by reducing the laser output power of AlSi12 deposition layer on the basis of sample 1. Finally, the cross-section of sample 2 was reduced to three regions and two interfaces, as shown in Figure 2d. The IMC region disappeared and was replaced by the white precipitate with a width of about 150 μm dispersed from the Nb/AlSi12 interface to the AlSi12 region (as shown in Figure 2e). Meanwhile, a good metallurgical combination was formed at the boundary between TC4 and Nb regions, with a smooth transition and no IMC formation, as shown in Figure 2f.

### 3.2. Phase Composition of Sample 1

The elemental distribution, phase composition and crystallographic information in sample 1 were further investigated using EBSD. As can be seen from the analysis results in Figure 3, Ti and Nb elements diffused in the entire region near the Nb/IMC interface, while the Al and Si elements separated in the upper region of the Nb/IMC interface. The massive Al-poor region was formed nearly 10 μm from the Nb/IMC interface, corresponding to the position of point 3 in Figure 2c. The EBSD phase scanning analysis (Figure 4) confirmed that the lower part of Nb/IMC interface was mainly composed of (Ti, Nb) solid solution and the Al-poor region was Ti_5_Si_3_ phase. The (Ti,Nb) solid solution solidified in the Nb region of sample 1 was melted by the larger AlSi12 deposition power. As a result, a large number of Ti, Nb, Al and Si atoms entered the molten pool simultaneously. In addition to Ti_5_Si_3_, γ-TiAl phase and NbAl_3_ phase, corresponding to points 2 and 4 in Figure 2c, were also formed in the upper part of Nb/IMC interface by the diffusion and migration between different atoms. The NbAl_3_, γ-TiAl and Ti_5_Si_3_ phases all contained either Ti atoms or Nb atoms at the same time (Table 3). This was because the atomic size and crystal structure of Ti and Nb are similar, resulting in the situation that Ti-Nb atoms occupied each other’s atomic positions in each phase [20]. The difference was that the content of occupying atoms was slightly different, but the basic lattice morphology of the phases was still maintained. Due to the existence of various compound phases, the structure of the deposited layer in sample 1 varied greatly, and it was easy to cause stress concentration and cracking under the action of cyclic thermal stress. In addition, the NbAl_3_ grains near the Nb/IMC interface were relatively coarse and had the same crystal orientation (growing along the (110) direction), as shown in Figure 3f. This morphology of the NbAl_3_ phase does not have enough anisotropic grain boundaries to hinder crack propagation. After crack generation, it extended along the IMC layer until the Nb/IMC interface. Finally, the initial crack was filled and healed by aluminum alloys with low melting point and good fluidity, while the distal part of the crack was not.

The EBSD analysis results at the IMC region and the IMC/AlSi12 interface are shown in Figure 5. The NbAl_3_ phase in the IMC region of sample 1 existed in the form of dendrites, as shown in Figure 5a. In sample 1, the deposited AlSi12 alloy and melted (Ti, Nb) solid solution made the Nb and Al composition contents in the molten pool (position of the blue line in Figure 6a) meet the atomic ratio requirements of NbAl_3_, and the NbAl_3_ crystal nuclei were precipitated directly in the liquid molten pool by homogeneous nucleation. After nucleation, Nb and Al elements in the molten pool diffused and migrated to the NbAl_3_ crystal nucleus simultaneously. In the appropriate crystal plane and crystal orientation, NbAl_3_ phase accumulated and grew into a primary dendrite axis, followed by very short secondary dendrites. With the formation of the NbAl_3_ phase, the content of Nb atoms in the liquid phase decreases, and the NbAl_3_ dendrites were filled and solidified to form α-Al solid solution. Figure 5d,e show the microstructure and phase distribution near the IMC/AlSi12 interface, which combines two states of NbAl_3_ phase in Figure 2b,e. The relatively loose NbAl_3_ phase was formed due to the lower Nb atomic content, and the Al content between NbAl_3_ dendrites was significantly higher than that in the IMC region.

### 3.3. Phase Composition of Sample 2

The XRD diffraction analysis results of the TC4 region, Nb region and AlSi12 region of sample 2 are shown in Figure 7. Combined with the EDS point scanning component analysis results in Table 3, it can be determined that the basket-like structure of the TC4 region was mainly composed of α-Ti, and β-Ti (the lower content made the diffraction peak not obvious) was present in the (α + β) organization. The AlSi12 region exhibited typical Al-Si eutectic characteristics accompanied by the massive primary Si phase. This is consistent with the research results of scholars Cheng et al. [21] and Gao et al. [22], respectively. The Nb region was mainly composed of (Ti, Nb) solid solution, and the white precipitate near the Nb/AlSi12 interface (as shown in Figure 2e) was NbAl_3_ IMC.

From the equilibrium state diagram of the Nb-Ti binary alloy (Figure 6b), it can be seen that an infinite solution between Nb and Ti can be achieved, and no IMC will be formed. When Nb was deposited on the surface of the TC4 deposition layer, Ti and Nb were thoroughly mixed in the molten pool by convection and the Marangoni effect [23]. As a result, a good metallurgical bond was formed at the TC4/Nb interface (Figure 2f), and the (Ti, Nb) solid solution with an atomic ratio of approximately 1:1 was also formed in the Nb region. Compared with the hard and brittle IMC generated by direct deposition of Ti-Al BS, the (Ti, Nb) solid solution has the characteristics of lower hardness and better plasticity, which can effectively relieve the stress concentration and reduce the crack sensitivity of the deposited layer to a certain extent. The composition analysis result of EDS point scanning in the Nb area (5 point position) was 46.76Ti-42.24Nb-8.52Al-0Si-2.48V (at%). The (Ti, Nb) solid solution of this composition (blue line in Figure 6b) has a relatively high melting point (~2000 °C) and completely transforms to the liquid phase at about 2100 °C. At lower AlSi12 deposition power, the (Ti, Nb) solid solution remained in the solid state, which effectively prevents the direct mixing of TC4 with AlSi12 and inhibited the formation of Ti-Al IMCs. At the same time, the dilution rate of the AlSi12 deposition layer was also reduced, which effectively inhibits the formation of NbAl_3_ and other IMCs. Different from Xu et al. [24] who used a niobium sheet to play the role of an obstacle in the laser welding process, the high melting point (Ti, Nb) solid solution formed after deposition plays a similar role to the non-melting niobium sheet.

As shown in Figure 2e, the white NbAl3 phase grew into the AlSi12 region in the form of columnar crystals along the surface of (Ti, Nb) solid solution. The formation of columnar crystals was due to the small temperature gradient and the fast solidification rate in the center of the molten pool. When the growth size was small, it appeared as granular or short rod-like with a smooth head, while the morphology of columnar crystal was maintained when the growth size was large. Strong internal stress tends to be generated in the preferentially grown columnar crystals. Moreover, the liquid metal in the molten pool undergoes vigorous stirring and convection under the action of the laser. The longer columnar crystals were easy to break into blocks due to the scouring of flowing metal and internal stress. Xue et al. [25] have recently shown that the path and distance of atomic diffusion and migration to a certain region were limited by diffusion time and temperature. As the distance to the Nb/AlSi12 interface increases, the ability of Nb and Ti atoms to diffuse and migrate gradually decreases, and the microstructure of pure AlSi12 alloy was formed after reaching a certain distance.

### 3.4. Fracture Mechanism of Sample 1

By comparing Figure 3f with Figure 5c,f, it can be seen that from the Nb/IMC interface to the IMC/AlSi12 interface, although the grain size of the NbAl3 phase became smaller, the crystal orientation was more homogeneous. However, the thicker NbAl_3_ IMC layer was always the weaker link in the BS organization compared to the more uniform and finer grain size of the aluminum alloy and the plastic (Ti, Nb) solid solution. The high thermal input of AlSi12 in sample 1 increased the reaction temperature and time, and formed a layered IMC region with a thickness of about 600 µm. The increase in the thickness of the reaction layer in turn adversely affect the component properties of the reaction layer [26,27,28]. During the constrained thermal contraction process, tensile stress was generated in each deposited layer. This eventually lead to more severe deformation and residual stress concentrations in the component, which increased the likelihood of cracking. Additionally, the research results of Li et al. [29] shows that the intrinsic brittleness of NbAl_3_ at room temperature increased with the increase of Ti doping concentration. As can be seen from Table 3, during the formation of NbAl_3_, Ti atoms occupied the position of Nb atoms and formed substituted solid solutions, thus further increasing the hardness and brittleness of the IMC region. Under the combined action of the above factors, a penetrating crack appeared locally in the IMC region of sample 1.

Although the formation of NbAl_3_ IMC in sample 2 could not be completely avoided by optimizing the process parameters, the brittle γ-TiAl phase and Ti5Si3 phase, which were more harmful to the structure, were eliminated. Meanwhile, the layered IMC was avoided and the microstructure variability of the deposited layers was reduced. In Specimen 2, NbAl3 dispersed in the AlSi12 region and the (Ti, Nb) solid solution with good toughness can significantly reduce the stress concentration. In addition, the timely repair of cracks by the aluminum alloy with good plasticity and mobility can prevent crack generation and expansion. It was proven that the formation of a high melting point (Ti, Nb) solid solution in TC4/AlSi12 BS with an Nb interlayer can effectively suppress the direct reaction of TC4 with AlSi12. Adjusting the heat input can effectively inhibit the formation of various IMCs and change the quantity, existence morphology and distribution of NbAl_3_ IMCs to avoid the formation of cracks.

### 3.5. Micro-Hardness and Tensile Properties

The microhardness distributions of the cross-sections of sample 1 and sample 2 are shown in Figure 8. As can be seen from the figure, there was no significant difference in the hardness of the TC4 region (about 340 Hv_x_) or the AlSi12 region (about 85 Hv_x_) in both samples, showing the highest and lowest hardness, respectively. A gradient transition with a hardness of about 200~220 Hv_x_ was formed between the TC4 region and the AlSi12 region. Obviously, the formation of (Ti, Nb) solid solution in the Nb region not only effectively hinders the interaction between TC4 and AlSi12 to form the high-hard Ti-Al phases, but also effectively alleviates the hardness difference between the two regions. The hardness of sample 2 gradually decreased from the TC4 region to the AlSi12 region, while sample 1 fluctuated between the Nb region and the AlSi12 region. This was due to the generation of a relatively hard and brittle NbAl_3_ IMC layer in this region, resulting in hardness exhibiting the second highest value of about 300 Hv_x_. The microhardness analysis results show that the Nb interlayer could obtain a good plastic transition from TC4 to AlSi12 under proper heat input control.

Three samples were prepared on sample 2 for the room temperature tensile test, and the tensile stress-strain curve was shown in Figure 9a. The experimental results showed that the tensile specimen fractured abruptly without a significant yielding with an average tensile strength of 128 MPa. The results indicate that TC4/AlSi12 BS with an Nb interlayer can obtain an acceptable tensile property under appropriate process parameters. The tensile fracture surface demonstrated mainly cleavage steps, showing the characteristics of quasi-cleavage fracture, as shown in Figure 9b,c. In order to determine the fracture position, an EDS point scanning analysis was performed on the fracture surface at point 1 and point 2 of the tensile sample in Figure 9c. It can be seen that the possible phases of the fracture were NbAl_3_ (Figure 9d) and (Ti, Nb) solid solutions (Figure 9e), indicating that the specific fracture location should be located near the Nb/AlSi12 interface of sample 2.

## 4. Conclusions

LAM technology was used to prepare TC4/AlSi12 BS samples with an Nb interlayer, which provided a new possibility for the preparation of a dissimilar material structure that was not limited to thin plate dissimilar welding only. The conclusions are summarized as follows:The accurate control of laser output power was crucial to obtaining TC4/AlSi12 BS. Higher laser power for the AlSi12 layer will produce a NbAl_3_ IMC layer with a thickness of about 600 μm.The TC4/AlSi12 BS prepared with optimized process parameters can be divided into three regions (the TC4 region, Nb region and AlSi12 region). The microstructure of the TC4 region was mainly composed of α-Ti and β-Ti, the Nb region was mainly composed of (Ti, Nb) solid solution, and the AlSi12 region was mainly composed of Al-Si eutectic and primary Si phase.The high melting point (Ti, Nb) solid solution formed after the addition of the Nb interlayer effectively prevented the reaction between the Ti and Al atoms. By optimizing the process parameters, the crack in the TC4/AlSi12 BS was successfully eliminated, the layered NbAl_3_ IMC was transformed into a dispersion distribution, and the interface in the cross-section was reduced from three to two.With the improvement of microstructure and stress conditions in the component, the tensile strength of TC4/AlSi12 BS with Nb interlayer was about 128 MPa.

## Figures and Tables

**Figure 1 materials-15-09071-f001:**
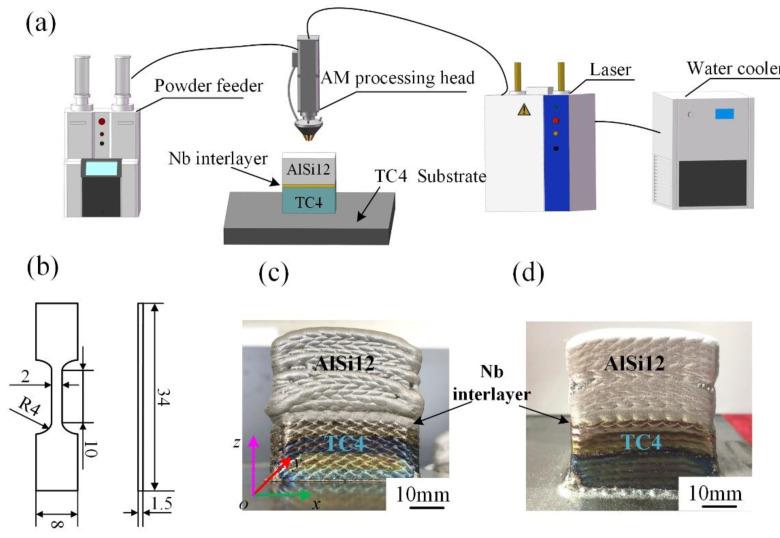
(**a**). Schematic diagram of LAM system, (**b**). Geometry and dimension of tensile specimen, (**c**,**d**). Physical photographs of sample 1 and 2.

**Figure 2 materials-15-09071-f002:**
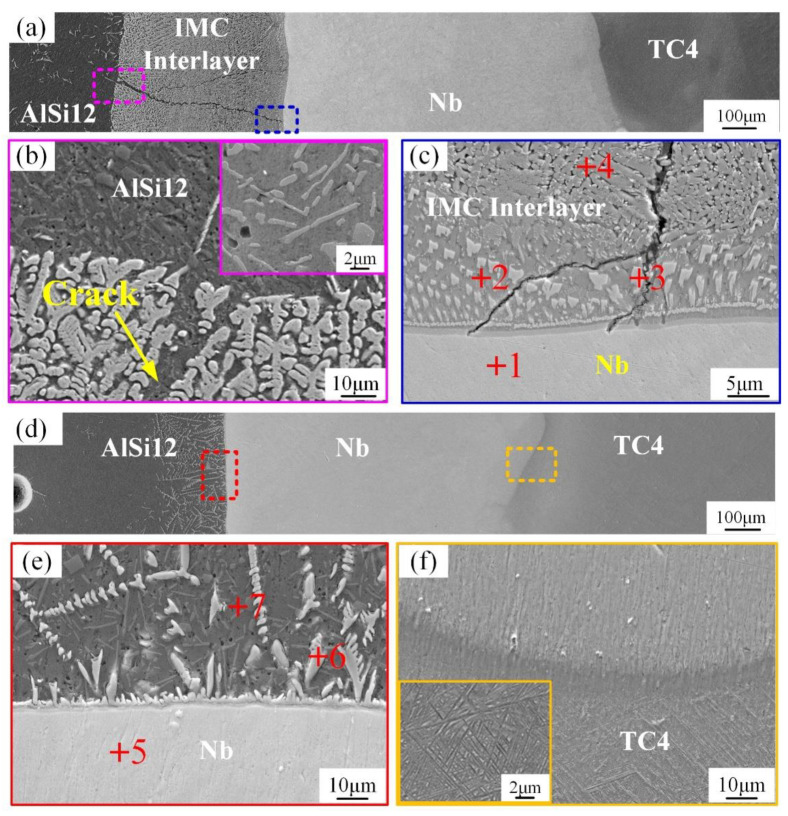
(**a**). cross-section of sample 1, (**b**). IMC/AlSi12 interface of sample 1, (**c**). Nb/IMC interface of sample 1, (**d**). cross-section of sample 2, (**e**). Nb/AlSi12 interface of sample 2, (**f**). TC4/Nb interface of sample 2.

**Figure 3 materials-15-09071-f003:**
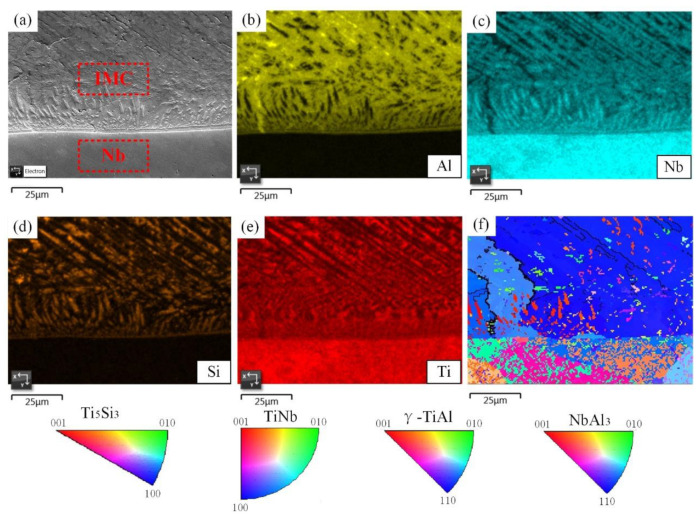
EBSD analysis results of Nb/IMC interface in sample 1, (**a**). SEM image, (**b**–**e**). EDS mappings, (**f**). IPF diagram.

**Figure 4 materials-15-09071-f004:**
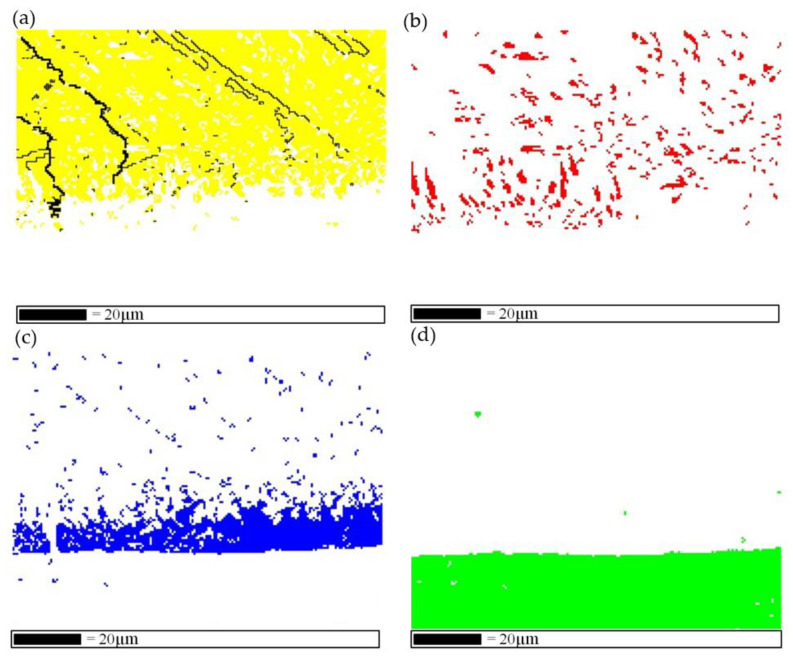
Phase distribution at the Nb/IMC interface in Figure 3a, (**a**). NbAl_3_, (**b**). Ti_5_Si_3_, (**c**). γ-TiAl, (**d**). (Ti, Nb) solid solution.

**Figure 5 materials-15-09071-f005:**
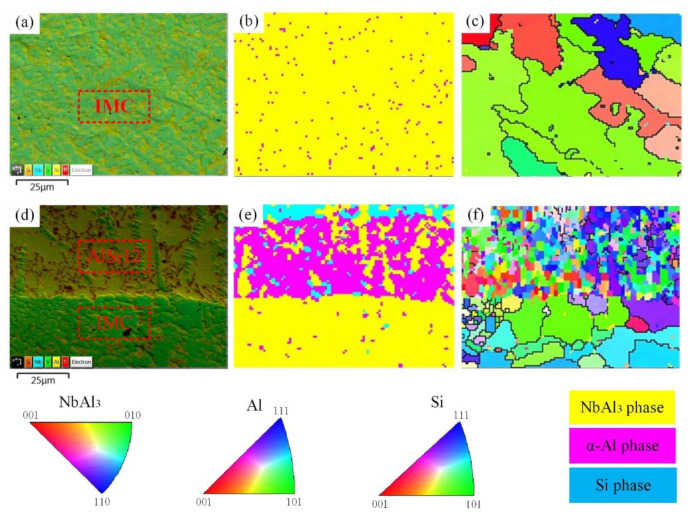
EBSD results of IMC region and IMC/AlSi12 interface in sample 1, (**a**). SEM image of IMC region, (**b**). phase distribution image of IMC region, (**c**). IPF image of IMC region, (**d**). SEM image of IMC/AlSi12 interface, (**e**). Phase distribution diagram of IMC/AlSi12 interface, (**f**). IPF diagram of IMC/AlSi12 interface.

**Figure 6 materials-15-09071-f006:**
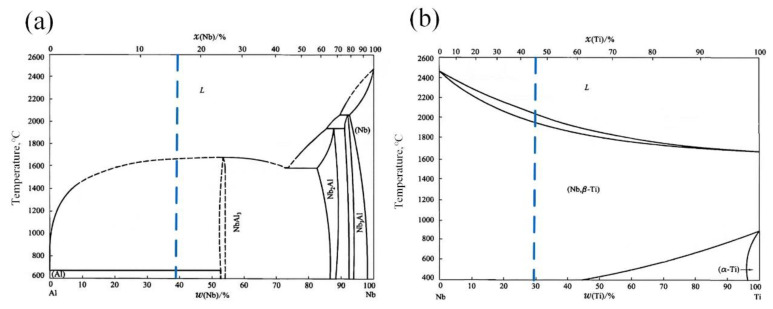
Equilibrium diagram of binary alloys, (**a**). Al-Nb, (**b**). Nb-Ti.

**Figure 7 materials-15-09071-f007:**
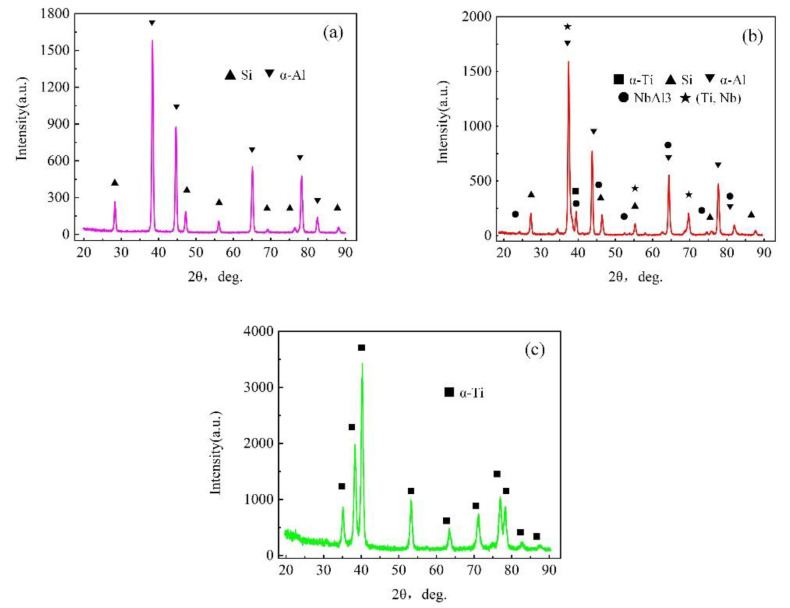
X-ray diffraction analysis results of different positions in sample 2, (**a**). AlSi12 region, (**b**). Nb region, (**c**). TC4 region.

**Figure 8 materials-15-09071-f008:**
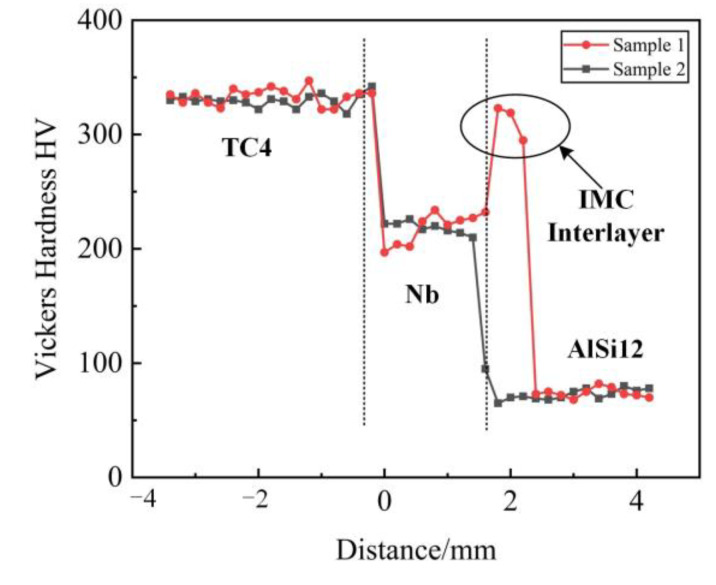
Microhardness distribution in cross-section of samples 1 and 2.

**Figure 9 materials-15-09071-f009:**
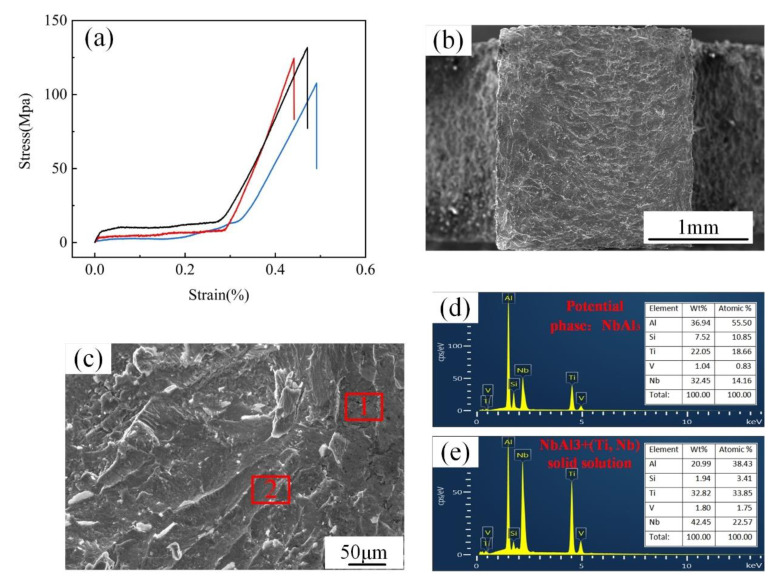
(**a**). Tensile stress-strain curve of sample 2, (**b**). Low magnification fracture morphology, (**c**). High magnification fracture morphology, (**d**,**e**). Component analysis results of point scanning at points 1 and 2, respectively.

**Table 1 materials-15-09071-t001:** The chemical compositions of TC4 and AlSi12 powders (wt%).

	Ti	Al	Si	V	Fe
TC4	Bal.	6.04	-	3.93	0.25
AlSi12	0.01	Bal.	11.96	-	0.18

**Table 2 materials-15-09071-t002:** LAM process parameters for the BS from TC4 to AlSi12 via Nb interlayer.

	Material	Laser Power (W)	Laser Scan Speed (mm/min)	Layer Thickness (mm)	Powder Feed Rate (g/min)
sample 1	TC4	2400	720	0.7	5.6
	Nb	2700	480	0.7	6
	AlSi12	1200	600	0.4	2
sample 2	TC4	2400	720	0.7	5.6
	Nb	2700	480	0.7	6
	AlSi12	1000	720	0.4	2

**Table 3 materials-15-09071-t003:** Spot EDS analysis results at different positions.

Location	Atomic Fraction of Element/%					Possible Phases
	Ti	Al	Nb	Si	V	
1	49.99	6.93	40.17	0.53	2.38	(Ti, Nb)
2	22.86	54.09	19.64	2.47	0.94	γ-TiAl
3	24.34	11.00	29.56	33.80	1.30	Ti_5_Si_3_
4	7.99	72.38	17.32	2.09	0.22	NbAl_3_
5	46.76	8.52	42.24	0	2.48	(Ti, Nb)
6	8.63	69.25	10.52	11.45	0.15	NbAl_3_ + Al
7	0.12	94.37	0	5.51	0	Al

## Data Availability

Not applicable.

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
