# Peer review of "Laser Additive Manufacturing of TC4/AlSi12 Bimetallic Structure via Nb Interlayer"

_materials, 2022, doi:10.3390/ma15249071_

Round 1
Reviewer 1 Report
The research is very interesting and the manuscript is well organized. Really titanium and alloy fusion welding is an actual challenge.
Some mistakes whiches need to correct,
In line 20: 128 MPa
Between the keywords, the „process parameters” in my opinion is not a keyword.
In line 29 find the first reference, but its number is [12]. It would be better in order.
In line 39 „methodsn” needs correction
The TiAl compound can be a gamma titanium aluminide? How did you determine this compound?
About the literature reference, this compound has excellent mechanical properties. How can it be the reason for the crack propagation?
Which kind of Ti-Al intermetallic compound can find in the IMC region?
In line 131 „was used to corrode the metallographic specimens”, maybe etched is a better expression for the metallographic preparation.
In line 310 the hardness correct form is 340 HVx (where x is the load) and also 85 HVx
Line 319 same HV problem, and sometimes (line 327) MPa designation is also wrong.
Line 328 dot before and after the reference.
Fig 9 d) and e), the subtitle is not clear.
In the conclusion, I can agree with conclusion 1, 3, and 4, but please increase the number 2 with the justification why are there only alfa Ti. May it find some gamma Ti?
The crack reason can be any gas contamination (ex. hydrogen attack).
Wish you good luck with the correction and waiting for your answers.
Author Response
Thank you very much for your comments concerning our manuscript. These valuable comments and sound suggestions are very helpful for revising and improving our manuscript. We have studied the comments carefully and responsed point-by-point. Please see the attachment.

Reviewer 2 Report
The paper is a contribution to investigation for producing method of the new multi-level materials known as the bimetallic composite structure based on Ti alloy/aluminium alloy. This dissimilar metal structural material is proving to be promising for the development of the aerospace industry, transportation etc. The use of classical technologies (welding, brazing) to fabricate such compositions has encountered many problems regarding development of their structure with a variety of intermetallic compounds which degrade the mechanical properties such as ductility and strength of weld and are reason for reduced fatigue properties.
In this paper, the authors present their research results in this area. To overcome the abovementioned issues, they applied the laser additive manufacturing technology with progressive idea to reduce the heterogeneity of structure using the Nb interlayer. The appropriate interlayer should allow to form no or less intermetallic compounds with the main metals to reduce crack sensitivity. To study the key microstructure characteristics the authors used the adequate advanced analytics methods.
Summing, the presented paper is characterized by good scientific quality but need some revision.
Reviewer comments:
11. I think, it would be proper to give information on the thickness of the Nb interlayer.
22. What the procedure was used to prepare the samples for tensile tests? Why the authors give the results of the tensile test only with the sample 2? What were the results for sample 1?
33. Why the authors used the different methods to analyse the phase composition of samples – for the sample 1 – EBDS and for sample 2 – XRD diffraction?
44. Line 357 – 359: Conclusions – point 4: Here, the authors state: “The improvement of microstructure and stress conditions in the component makes the tensile strength of TC4/AlSi12 BS with Nb interlayer increased by about 15% compared with TC4/AlSi12 BS without Nb interlayer, reaching about 128Mpa.” I think, here, the reference of [26] should be given because no information on the tensile strength of TC4/AlSi12 BS without Nb interlayer are given.
Minor comments:
11. In the “Keywords”, I recommend substituting the acronyms such as “BS” or “IMCs” with the full words “bimetallic structure” or “intermetallic compounds” because the reader usually doesn't look for acronyms as keywords.
22. The acronyms such as “BS” or “IMCs” should also be introduced in the main text not only in the abstract.
33. Line 125-126: There is written “… Fig 1 (d) … resulting in a flatter and smoother surface and sides.” In Fig. 1 (d), the surface is cut off.
44. In the subchapter 3.2., there is missing the reference on Fig. 4.
55. Line 148: In the caption of Fig. 1, there is written “… Geometry and dimension of tensile dimension …”.
66. Line 122 – 123: I think, at the end of sentence, the reference to [26] could be given.
Author Response
Thank you very much for your comments concerning our manuscript. These valuable comments and sound suggestions are very helpful for revising and improving our manuscript. We have studied the comments carefully and responded point-by-point. Please see the attachment.

Reviewer 3 Report
The present work entitled "Laser additive manufacturing of TC4/AlSi12 bimetallic structure via Nb interlayer" describes the influence of Nb interlayer on the structure and mechanical properties in the TC4/AlSi12 bimetallic structure joint.
In the introduction to the work, the advantages of additive technologies and the need to use a Nb barrier layer are described in sufficient detail.
However, it is not entirely clear for what specific purposes the resulting bimetallic composite can be used. Based on this, it is not clear what requirements for properties are imposed on this composite.
Suggestion 1: Specify in the introduction what properties the resulting composite should have. And what is the function of the TC4 and the AlSi12 layers?
The introduction does not substantiate the choice of LAM technology. Other methods for producing bimetallic structures, such as surfacing, cold gas-dynamic spraying, explosion welding, are not considered. Some methods make it possible to obtain high-quality joints with the almost complete absence of diffusion interaction between the components, which significantly increases the tensile strength.
Suggestion 2: Expand the introduction by considering other technologies for obtaining bimetallic structures and justifying the choice of LAM technology.
It is not entirely clear what size layers were applied. The described data on 40x25x20 “cuboids” (lines 119-121) do not quite correspond to the scale of Figure 1, d. There is no information on the thickness of the Nb layer.
Suggestion 3: Specify the thickness of the applied layers.
It is not clear from methods how the samples were prepared for X-ray diffraction analysis. How was the Nb region analyzed (fig. 7b)? X-ray diffraction studies were supposed to confirm the presence of intermetallic phases (NbAl3 and etc.) in the AlSi12 layer and Nb/AlSi12 interface (fig. 7a).
Suggestion 4: Describe the preparation of samples for X-ray diffraction analysis. Justify the absence of intermetallic phases in the diffraction pattern of the AlSi12 region (fig. 7a).
When applying Nb layer, active remelting occurs with the formation of a solid solution of Nb and Ti. Why were such laser parameters chosen for Nb layer deposition? It is possible that the absence of Ti in the Nb layer can further increase the strength of the composition.
Suggestion 5: Justify the choice of LAM process parameters for Nb layer deposition.
The content of almost 50% Ti in the Nb region raises questions about the effectiveness of preventing the reaction between Ti and Al. Perhaps X-ray diffraction analysis will reveal the presence of Ti-Al phases at the Nb/AlSi12 interface.
The authors note that the tensile strength of TC4/AlSi12 BS with Nb interlayer increased by 15% compared to TC4/AlSi12 BS without Nb interlayer [26]. However, in Ref. [26] AlSi12 layer was deposited on TC4 with different LAM process parameters. Therefore, such a comparison is incorrect. It is possible that laser deposition modes have a more significant effect on the strength of the composition than the presence of Nb layer.
Suggestion 6: Clarify the effect of deposition parameters on the tensile strength.
The paper with opportune modifications can be presented to reconsideration.
Author Response

(The authors gave the same response as above.)

Round 2
Reviewer 3 Report
The authors corrected the manuscript according to some of the reviewer's comments. In general, the article can be accepted for publication.